# OpenReview forum: "OMNI-LEAK: Orchestrator Multi-Agent Network Induced Data Leakage"
_ICLR.cc/2026/Conference — ICLR 2026 Conference Desk Rejected Submission_

### Official Review · Reviewer_XCKC · 2025-10-29

**Soundness:** 2
**Presentation:** 3
**Contribution:** 2
**Rating:** 4
**Confidence:** 3

**Summary:**

The authors introduce an attack on Multi-Agent Systems called OMNI-LEAK, which compromises several agents to leak sensitive data through a single indirect prompt injection, even in the presence of data access control.


The attack is measured on several frontier models finding that both reasoning and non-reasoning models are vulnerable, even when the attacker lacks insider knowledge of the implementation details. Our work highlights the failure of safety research to generalize from single-agent to multi-agent settings, indicating the serious risks of real-world privacy breaches and financial loss.

**Strengths:**

The paper is well-written, clear and addresses an important vulnerability in data leakage within agents and multi-agent systems.

**Weaknesses:**

- **Novelty:** The original control flow hijacking paper demonstrates data leakage (using Python and online input vs. SQL in this paper). Outside of changing the language and the input modality, I am struggling to see how this paper differentiates itself from the original. At the very least CFH should be used as a baseline in this work.
- **Evaluations:** While any instance of data leakage is potentially catastrophic, in some of the authors' experiments, the attack does not succeed and in others takes several (500) tries to succeed.
- **Defenses:** Some defense like [this one](https://arxiv.org/html/2502.01822v1) are designed, in part, for this vulnerability. These should be evaluated against.

**Questions:**

See above

---

> ### Author Response · Authors · 2025-11-22
> **Rebuttal to XCKC**
>
> We would like to thank the reviewer XCKC for their helpful comments and suggestions. In particular, we are thankful to the reviewer for highlighting our contribution in addressing a novel vulnerability in multi-agent systems. Below, we respond to other comments and questions raised by the reviewer:
>
> W1:
>
>
> Indeed, our work shares similarities with the concurrent work of [Triedman et al. 2025](https://openreview.net/forum?id=DAozI4etUp#discussion), which was published only a few months before the submission deadline.
>
> Our paper is complementary to theirs, as while there are similar mechanics used in the attack, the details of the setting being considered are different, and importantly, their analysis does not consider access control. As stated on Line 124, their paper focuses on code execution rather than the leakage of private data. The major contribution of our work is to show a simple method to leak data in a multi-agent system with access control, without the need for agents to be directed to compromised documents or webpages. Access control has long been one of the main ways of keeping data secure. To the best of our knowledge, we are the first to show how access control can be bypassed using prompt injection attacks and compromised agent-to-agent interactions. Our attack vector can be used in many systems where the work of Triedman et al. 2025 would not be appropriate and would require unrealistic access assumptions, such as in-house data management tools without web or directory access.
>
> W2:
>
> Regarding the evaluation, we are unsure why “some models are harder to attack than others” is a weakness of our work. Conversely, we think this is one of the most interesting findings and shows that different AI companies' approaches to AI safety are likely to have different results. We note that Triedman et al. 2025 did not present any results for a model developed by Anthropic, about which we provide a further detailed analysis in Appendix E. As the reviewer acknowledges, with data leaking attacks, a high attack success rate isn’t necessary to cause major harm, as the data only needs to be leaked once, the attack has maximum effect.
>
> W3:
>
> While we agree that there is a lot of interesting work to be done in the space of defending against prompt injection attacks and comparing the efficiency of different techniques in different settings. This sort of analysis is out of scope for our paper, which presents a new attack vector. We note that many accepted attack papers do not offer results against recent defence mechanisms.

---

### Official Review · Reviewer_uFkU · 2025-10-29

**Soundness:** 2
**Presentation:** 3
**Contribution:** 2
**Rating:** 4
**Confidence:** 3

**Summary:**

The authors designed an attack method named OMNI-LEAK over an industrial multi-agent pattern named orchestrator to induce multiple agents to leak information through a pervasive approach. Attackers need only inject once for leaks to propagate across layers. The authors argue that even if each submodel is individually secure, collaborative modes may introduce new attack surfaces. Multi-agent data leakage is a structural issue: as long as information is conveyed in natural language among agents, attackers can exploit contextual reasoning to achieve leakage.

**Strengths:**

1. This paper is closely integrated with the industry, employing a comprehensive orchestrator framework. Addressing multi-agent data leakage is of broad relevance to the community.
2. The details of OMNI-LEAK prompt injection are described very clearly and are easy to understand.
3. The verizon that even if each agent is secure individually, the combination may still lead to emergent vulnerabilities, is of great insight.
4. The article provides numerous detailed steps for reproduction, making it highly reproducible.

**Weaknesses:**

1. The paper leans towards engineering, with insufficient attack formulations presented. The procedures lack formulaic standardization.
2. The paper lacks analysis and comparison with other attack methods, such as direct prompt injection and jailbreak attacks. This would better highlight OMNI-LEAK's advantages, enabling deeper analysis—whether in terms of robustness or the expected number of queries required for a successful attack.
3. The insight that even if each agent is individually secure, their composition can still yield emergent vulnerabilities is particularly compelling, revealing a structural weakness in collaborative agent systems.
4. The experimental design was overly simplistic, lacking diverse baselines and rich metrics, and failed to yield findings with insight or generalizability.

**Questions:**

1. Please provide some symbolic or theoretic formalization of their attack chain? It seems this would be useful both to try to generalize results, and to reason about other orchestrator-based workflows.
2. Please quantify the magnitude of leakage (e.g., number of leaked tokens, sensitivity levels) to compare with different types of attacks? I believe this study can enhance the technical quality.
3. Please include analysis for direct prompt injection and jailbreak, as well as the defense methods of 10 kinds of indirect prompt injection attacks.
4. The paper assumes free-form natural-language passing via an orchestrator. Must this communication mechanism be free-form to enable OMNI-LEAK, or would  schema or parameterized calls prevent the layered propagation? Please clarify.

**Details Of Ethics Concerns:**

The paper contains benchmarking elements, and the author should make the code publicly available to verify whether the data contains any privacy leaks. Even without such checks, the code itself should be made publicly available.

---

> ### Author Response · Authors · 2025-11-22
> **Rebuttal to uFkU**
>
> We would like to thank the reviewer uFkU for their helpful comments and suggestions. In particular, we are thankful that the reviewer highlighted the importance of understanding how individually secure agents, when composed, can give rise to emergent vulnerabilities. This observation captures the core motivation of OMNI-LEAK. Below, we respond to the remaining comments and questions raised by the reviewer:
>
> W1 and Q1:
>
> We have now added an appendix formalizing the multi-agent orchestrator system, along with each of its components. To highlight briefly, we have formalized the Orchestrator Agent, Data Processing Agent, Messenger Agent, and an extensible set of optional sub-agents. Further, we have formalized the adversarial objective as well. Please consult Appendix F (in blue) in our revised paper.
>
> W2 and Q3:
>
> The comparison of indirect prompt injection to direct prompt injection and jailbreak attacks is an important one. In Section 2, we differentiate traditional prompt injection (direct prompt injection) from indirect prompt injections, and further clarify it in our orchestrator design (Section 4) to highlight the uniqueness of the attack with respect to jailbreaking and prompt injection: “Because the SQL agent is physically unable to retrieve the private data when the adversary is directly using the orchestrator system, any form of jailbreaking or direct prompt injection attacks fails.”
>
> Traditional prompt injection/jailbreak attacks rely on manipulating input instructions, which can be gated with scanners or role-based access control. By contrast, OMNI-LEAK exploits emergent interactions between agents, where one agent summarises or transforms private data into a benign-looking natural-language form that bypasses privilege boundaries. Scanners that prevent direct input/output bypasses are ineffective here because the leakage occurs during legitimate intermediate agent–agent communication, not through direct user prompts.
>
> When an unprivileged user interacts with the orchestrator, the SQL agent cannot access private data or even detect its existence. Therefore, any direct prompt injection or jailbreak attempts by an unprivileged adversary fail, as the downstream agent cannot retrieve confidential rows.
>
> There is a wide encapsulation for employing this strategy, but no standard procedure for automatically constructing these indirect prompt injections. We may revise these sections to make this distinction even more explicit (and we’re open to suggestions on how to do so!), as we recognise that this clarification is essential to interpreting our results.
>
> W3:
>
> We thank the reviewer for recognizing this insight—indeed, our key finding is that agent composition introduces new, systemic vulnerabilities even when each component agent individually satisfies security constraints. We did not feel this was considered a weakness in our work, so if the reviewer would please elaborate on what they see as a weakness here that would be appreciated?
>
> W4:
>
> Our goal in this initial study was to highlight the emerging attack surface introduced by agent composition, independent of application-specific complexities. We utilize the expected number of queries required for a successful attack, $\mathbb{E}_\text{attacks}$, as our main metric to measure attack success, which is equivalent to $\lceil \frac{1}{ASR} \rceil$, where ASR is a common (and sometimes the only) metric used in many red-teaming papers. In addition, we also include benign query accuracy and robust benign query accuracy as additional metrics. Can the reviewer please recommend missing metrics or baselines?
>
> Q2:
>
> We agree with the reviewer about expanding our particular leakage metrics, and we expand the evaluation of OMNI-LEAK itself by quantifying the leakage:
> - Leaking 5 / 20 / 100 SSNs under escalating attack complexity
> - Each SSN consists of 11 characters (9 digits + 2 dashes)
>
> It should be noted that OMNI-LEAK supports arbitrary payload sizes and formats; the leaked information depends on the specific database(s) and summarisation behaviour. We have included these metrics explicitly (in blue) in the revised manuscript as part of Appendix E.

---

> > ### Author Response · Authors · 2025-11-22
> > **Rebuttal to uFkU (continued)**
> >
> > Q4:
> >
> > We appreciate this insightful question. OMNI-LEAK does not fundamentally require free-form communication, but free-form agent hand-offs greatly increase the expressiveness and the interpretability of the system to modify its behavior as a result. Whether these instructions are returned as free-forming text or encapsulated within a schema or parametrized call, the instructions would remain in the original text entry field. The Orchestrator Agent LLM’s inability to distinguish between informational context and actionable instructions, and their lack of awareness in avoiding the execution of instructions within external content, would remain valid ([Yi et al, 2025](https://dl.acm.org/doi/abs/10.1145/3690624.3709179)).
> >
> > In our evaluated setting:
> >
> > - Even under more structured schemas, many orchestration frameworks include at least one text field or descriptive parameter, which can carry sensitive information.
> > - Restricting fully parameterized calls may mitigate, but does not eliminate the attack surface, as long as any agent is allowed to re-encode private data into natural language.
> > - The SQL setup has access to this database, so there would be some point in the schema where there is an input field, because this input field may be a malicious instruction.
> > - Since it is widely known that agents cannot differentiate between instructions and data, and since SQL agents load data by design. Whether it is loaded in free form or a parameterized set, this data could be maliciously executed if it were presented as instructions.

---

### Official Review · Reviewer_eJQb · 2025-10-31

**Soundness:** 1
**Presentation:** 3
**Contribution:** 1
**Rating:** 2
**Confidence:** 3

**Summary:**

This paper proposes a prompt injection attack within an orchestrator multi-agent setup, in which hidden malicious instructions are injected into public data to induce an SQL agent to leak private information. A benchmark is designed to evaluate the vulnerability of various LLMs against the proposed attack.

**Strengths:**

S1. The writing is clear and easy to follow.

S2. Currently, there is limited research on the security of multi-agent LLM systems, and the authors are trying to contribute to a promising emerging direction.

**Weaknesses:**

W1. Although the paper claims to propose a novel data leakage attack that compromises multiple agents, the method just employs conventional hidden prompt injection attacks to manipulate the SQL agent for unauthorized data access. I do not believe this approach represents a significant advancement over existing work. It is unclear to me what new defensive challenges are introduced by the attack proposed in this paper. It appears that existing defenses against indirect prompt injection attacks [a] could already mitigate the proposed attack.

[a] Can Indirect Prompt Injection Attacks Be Detected and Removed? ACL 2025.

W2. The paper does not discuss any potential defense mechanisms.

W3. The proposed benchmark does not incorporate any defense mechanisms, and therefore, I do not believe it appropriately reflects real-world attack challenges.

W4. The evaluations consider only data scenarios within the employees/HR domain, which raises concerns about the generalizability of the experimental findings.

W5. The paper claims that the proposed method can be extended to other orchestrator setups, but I did not find any detailed discussion or supporting evidence.

**Questions:**

Q1. Compared to indirect prompt injection attacks, what new defensive challenges does the proposed attack introduce?

---

> ### Author Response · Authors · 2025-11-22
> **Rebuttal to eJQb**
>
> We would like to thank the reviewer eJQb for their helpful comments and suggestions. In particular, we are thankful to the reviewer for highlighting our contribution in the newly emerging field of multi-agent security. Below, we respond to other comments and questions raised by the reviewer:
>
> W1:
>
> We do not claim our contribution comes from innovating a new prompt injection attack methodology. Our novel contribution comes from showing how to use compromised agent-to-agent interactions to leak data in a realistic multi-agent system with access control. Access control is one of the main defence mechanisms used to keep data secure in the context of data management. To the best of our knowledge, we are the first paper showing how access control can be bypassed using prompt injection attacks and compromised agent-to-agent interactions. We show how this attack can be applied in a wide range of realistic settings, following real-world design patterns. Our attack vector can be used in many systems where standard environmental attacks would require unrealistic access assumptions, such as in-house tools developed by companies without web access.
>
> Thanks for sharing this interesting paper [a]. This paper presents a method for detecting prompt injection attacks in documents. Yes, running this sort of system as a filter on all user requests and data entering the multi-agent system could likely remove some attacks. This would increase the computational overhead and could lead to high false positives, where the system fails to execute benign instructions, which would make the system frustrating to use. However, our paper is attack focused showing how to bypass access control in particular not other recent defences published a few months before the submission date.
>
> W2 and W3:
>
> Our attack incorporates perhaps the most widespread defence mechanism in the context of data management, which is access control. To the best of our knowledge, we are the first paper that shows how this ubiquitous defence mechanism can be defeated in a number of realistic scenarios.
>
> Further, we have now added a list of defensive recommendations as per the suggestion of reviewer bo4i. Please read the the updated conclusion in our paper (in blue).
>
> W4:
>
> In preliminary work, we experimented with three different databases representative of different use cases. These included a healthcare database containing patient IDs, diagnoses and insurance numbers, and a transaction database including transactions, merchant IDs and credit card numbers.  However, in our initial experiments, these settings showed little difference in behavior. Hence, to reduce the computational and space requirements in the paper, we selected one of these as a representative example of the attack. Our attack should be effective on all settings that can be represented by the formalism described in the response to reviewer uFkU.
>
> W5:
>
> We report results over five different orchestrator setups with the number of worker agents varying from 1 to 4, powered by 5 different LLM models, both reasoning and non-reasoning, being used in different configurations. In nearly all settings, we find attacks that are successful. To us, this seems indicative that OMNI-LEAK generalises across orchestrator setups. Could the reviewer explain what evidence they think is lacking?
>
> Q1:
>
> Our attack highlights two main new defensive challenges to indirect prompt injection attacks.
>
> First, we show that including a SQL database as part of a multi-agent system presents a new vector for both prompt injection attacks to be ingested, but also for data to be exfiltrated. We show that access control, which has typically been used to protect against data leakage, is insufficient in the era of LLM-powered agents. To the best of our knowledge, we are the first work demonstrating this.
>
> Additionally, we exhibit that orchestrator setups present a number of unique defensive challenges, over and above indirect prompt injection attacks on single-agent systems.  For example, in section 5.2, we demonstrate that once attacks have hijacked a worker agent, they become increasingly difficult for other agents to detect. Meaning even LLMs that demonstrate a high level of security as a worker agent do not provide good security when acting as an orchestrator, if the worker agent(s) are compromised.  Further, if we compare to the user-single agent case, the user can identify when the agent starts going wrong when it outputs suspicious text. But in the case of multi-agent systems, any potential suspicious activity by the worker agents is abstracted away from the user, with the orchestrator in between.

---

> > ### Comment · Reviewer_eJQb · 2025-11-22
> >
> > Thank the authors for their efforts in clarifying important concerns. I provide my response as follows.
> >
> > > "However, in our initial experiments, these settings showed little difference in behavior. Hence, to reduce the computational and space requirements in the paper, we selected one of these as a representative example of the attack."
> >
> > You should include these experimental results in the paper to support your claims. I do not think the page limit is an issue, because you can put these results in the appendix. Moreover, I am not sure what these initial experiments were like. Even if the results of these setups did not differ significantly in the initial experiments, that does not mean the same will hold in the subsequent, more in-depth experiments.
> >
> > > "To us, this seems indicative that OMNI-LEAK generalises across orchestrator setups. Could the reviewer explain what evidence they think is lacking?"
> >
> > In the Introduction, the authors state that "To this end, we incorporate a Data Processing Agent using SQL in all orchestrator setups we examine, although our red-teaming strategy is agnostic to the specifics of the SQL Agent and can be extended to other orchestrator setups." When I read this sentence, I thought that "other orchestrator setups" referred to setups other than the "orchestrator setups we examine" in Section 5.3. This may have been a misunderstanding; the authors may have simply intended to say that they also evaluated three additional setups in Section 5.3.
> >
> > My concern stems from the fact that this paper has not discussed how the four orchestrator setups are sufficiently representative of the various complex setups in the real world, nor how the paper’s approach can be extended to setups that have not been examined in the experiments.

---

> > > ### Author Response · Authors · 2025-12-02
> > >
> > > We thank the reviewer eJQb for their rapid response. We have now added the preliminary results in the appendices. While we agree that subsequent, in-depth experiments could reveal differences between the employees/HR database and the orchestrator setups we examined, we wanted to focus more on the different orchestrator setups, given our limited compute budget and our preliminary experiments. Doing all our experiments on N different databases would increase the compute needed by a factor of N.
> > >
> > > Moving on to the next comment, we appreciate the reviewer pointing out where the confusion stems from. We will amend the language in the introduction to be more specific to orchestrator settings where we have evidence of generalization, particularly ones including a data processing agent (such as SQL agent).

---

### Official Review · Reviewer_6R2i · 2025-11-01

**Soundness:** 3
**Presentation:** 2
**Contribution:** 2
**Rating:** 6
**Confidence:** 3

**Summary:**

The paper OMNI-LEAK: Orchestrator Multi-Agent Network Induced Data Leakage investigates security vulnerabilities in multi-agent systems, particularly within the orchestrator setup where a central agent delegates tasks to specialized agents. The authors present a novel attack vector, OMNI-LEAK, which exploits indirect prompt injections to compromise multiple agents and leak sensitive data, even when data access control safeguards are in place. The study demonstrates that various state-of-the-art language models, except for Claude-Sonnet-4, are vulnerable to such attacks and highlights the importance of considering multi-agent vulnerabilities in security research.

**Strengths:**

1. The paper introduces a new attack vector, OMNI-LEAK, and addresses the gap in research on multi-agent system vulnerabilities, which is a critical area for ensuring the safety of modern AI applications.

2. The authors provide comprehensive experiments across different models, databases, and attack categories, offering a detailed analysis of model susceptibility to OMNI-LEAK.

3. The study uses practical examples, such as employee database management, to demonstrate the vulnerability of multi-agent systems, making the findings highly relevant for both academia and industry.

**Weaknesses:**

1. The beginning of Chapter 4 introduces the experimental setup, but there is no clear justification for why this particular setup was chosen. The authors should provide a more robust explanation of the rationale behind the design of the system. This could be done through theoretical reasoning or supported by practical insights, such as findings from similar studies or real-world surveys. Without this, the setup appears arbitrary and weakens the foundation of the paper.

2. While terms such as "SQL agent" are used frequently, their definitions are not fully explained. As different researchers may interpret such terms differently, it is important to align the terminology used and provide clear definitions. For instance, the role and functionality of the SQL agent should be clearly stated to avoid confusion and ensure that readers can follow the arguments easily.

3. The paper frequently mentions the Claude-Sonnet-4 model’s robustness in dealing with attacks, but there is a lack of in-depth analysis of why this model demonstrates such robustness. A more thorough exploration into the characteristics, design decisions, or training methods that contribute to its unique resistance to attacks would strengthen the paper's contribution and provide useful insights for both security researchers and AI model developers.

**Questions:**

See weakness

---

> ### Author Response · Authors · 2025-11-22
> **Rebuttal to 6R2i**
>
> We would like to thank the reviewer 6R2i for their helpful comments and suggestions. In particular, we are thankful to the reviewer for highlighting our new attack vector OMNI-LEAK, and our comprehensive experiments. Below, we respond to other comments and questions raised by the reviewer:
>
> W1:
>
> We respectfully disagree with the reviewer that there is no clear justification for why the particular setup was chosen, although it wasn’t all in Section 4. Most of our motivation for choosing this setup is highlighted in Sections 1-3, particularly in our Introduction. Several companies, such as Google, Microsoft, Accelirate, Adobe, AWS, IBM, EmergenceAI and ServiceNow, provide agentic orchestrator system services (referenced in lines 53-54 and 62-63 on page 1 of our paper). Further, many of them support data management as a common use case that has a clear and concerning threat of data leakage. We further motivate our choice of Notification Agent in Section 4, lines 189-196. Our experimental setup represents one key instance of a setup involving an orchestrator, data agent and messaging agent. An SQL agent can easily be substituted by a RAG agent, another memory or data retrieval agent, while the email in the Notification agent can easily be substituted with SMS, Slack or other kinds of notification or alerting systems.
>
> Moreover, we would like to point out that there is little research done in this direction of agent-to-agent failure due to injections. To our knowledge, there is a severe lack of comprehensive studies or surveys that detail the real-world usage of multi-agent systems, let alone orchestrator setups in particular. This is mostly due to the fact that this is still an emerging technology that the industry has started to deploy.
>
> W2:
>
> We have loosely defined the SQL agent in Section 4, lines 153-156. However, for completeness, we now present a more formal definition of SQL agent as follows: an SQL agent is an agent that takes in a natural language query P, generates an SQL query Q that helps answer P, and then returns a natural language response R, supported by the information gathered from running Q. To tie in to our existing Figure 3, P can be “What department does Mark work in?”, Q can be “SELECT * FROM employees WHERE first_name = ‘Mark’;”, and R can be “Mark works in Engineering”.
>
> We have added this to the paper under Section 4 (in blue). We are happy to add further clarifications as per the reviewer.
>
> W3:
>
> We have already done a deeper analysis of claude-sonnet-4 which is reported in Appendix E. This is why we wrote a short Appendix E discussing the potential reasons that contributed to claude-sonnet-4’s robustness. We hypothesise that claude-sonnet-4 may have undergone safety fine-tuning that improved its ability to detect phishing attempts involving suspicious emails.
>
> If there is any additional analysis the reviewer believes could be useful for us to do, given the publicly available info on claude-sonnet-4’s safety training, then please feel free to tell us.

---

### Official Review · Reviewer_bo4i · 2025-11-01

**Soundness:** 2
**Presentation:** 3
**Contribution:** 2
**Rating:** 4
**Confidence:** 3

**Summary:**

Paper presents study of large language model (LLM) -based orchestrated multi-agent system risks for security vulnerabilities of leaking private data with indirect prompt injections. A novel attack vector is presented and experimented in office employee/HR use case with several public LLM-based agents with quantitative analyses. As a results both reasoning and non-reasoning LLMs are vulnerable to proposed attacks in the networked agent setup.

**Strengths:**

The topic is important and timely. To my knowledge, studying of the coordinated multiple LLM agent setup for data leakage with prompt injection attacks, is new. Paper is clearly written and has good illustration of the problem and attack executions. In the specific benchmark, different aspects (target models, database sizes, attack categories and level of information accessed) are evaluated by measuring the statistics of query accuracy and successful attacks, given some interesting findings where almost all models are vulnerable for these specific attacks of leaking the private data. In high level this might provide valuable information for the community.

Summary of the strengths
- Important and timely problem
- Novel multi-agent prompt attack setup and benchmark
- Benchmark evaluation gives some interesting findings how easily the models leak the data

**Weaknesses:**

Although in high-level the study setup and benchmark is well-defined, there are some limitations. The main weaknesses are related to the lack of detailed analysis of the models, decreasing the significance and quality of the contributions. Now, the concluding remarks are quite speculative, and the reader is left with the question of why certain models are more or less vulnerable. Is it the fine-tuning or some other aspect of the model or training data effecting the results. From these perspectives, if possible with these black-box model, it would be good to analyse more detailed the effect of fine-tuning and other constrains (e.g. RAGs) to the success of attacks. Also, based on the analysis it would be good prepare a summary or list of actionable item for the practitioners of how to utilise the findings.

Summary of weaknesses
- Limited and speculative reasoning and analysis of the particular models; what are the characteristics why certain (black box) model is more or less vulnerable
- Limited analysis of the details of the models and what would be the effects of possible fine-tuning/RAG etc. to success of the injection
- Missing a summary/list of actionable items for engineers/practitioners of how to utilise the findings

**Questions:**

- The concluded analyses of the attack success considering the target models are quite speculative. Are there any possible additional techniques that could be used to analyse what are the common characteristics or patterns or what are the properties of certain LLM leading to worse or better protection of the attacks?
- Are there any summary (of lesson learned) for practitioners and engineers in industry of how to prepare for these kind of vulnerabilities?

---

> ### Author Response · Authors · 2025-11-22
> **Rebuttal to bo4i**
>
> We would like to thank the reviewer bo4i for their helpful comments and suggestions. In particular, we are thankful to the reviewer for highlighting the novelty of our multi-agent attack setup and why it is so pressing to red-team it. Below, we respond to other comments and questions raised by the reviewer:
>
> W1 and Q1:
>
> We would like to clarify that we see our paper’s major contribution as demonstrating the susceptibility of frontier models to this novel and unique agent-to-agent failure. We do not claim that this paper is about understanding the reasoning behind why certain models are more vulnerable than others. However, this is something we thought about, but the majority of models tested were closed-source APIs (as those are the most capable models). The lack of information on the exact safety finetuning, datasets, adversarial safety training, and safety monitors makes it difficult to compare the models on the specifics of their safety training.
>
> Moreover, this should not be considered a limitation to our work, as this is a general limitation associated with using closed-source models. Several recent red-teaming papers have been published in tier one venues that do not provide compelling reasons that distinguish the model’s safety properties. For example, see [Pedro et al. (2025)](https://ieeexplore.ieee.org/document/11029790), [Jones et al. (2024)](https://openreview.net/forum?id=GCkhEPE1FG),  [Greshake et. al (2023)](https://dl.acm.org/doi/10.1145/3605764.3623985) cited in Sections 2-3 in our paper. If the reviewer has suggestions on what experiments we could run to uncover this information, we’d be happy to listen to them.
>
> W2:
>
> We are not clear what the weakness pointed out is; is the reviewer suggesting fine-tuning on these injections as a defence mechanism when they say “fine-tuning to better success of the attack”?
>
> It is possible that finetuning against the exact attacks that we used would make a given model more robust against those attacks. However, these are only a subset of multi-agent attacks in a particular employee/HR domain, whose purpose was only to highlight the vulnerability of these models. Finetuning against only this subset may not generalise to the larger category of attacks. Extensive adversarial training (such as automating the red-teaming via an LLM and continually finetuning on it) may grant better protection. However, this is a substantial endeavour that is best left to future work, along with investigations into other defence mechanisms such as tagging data from instructions.
>
> As for the effects of RAG, is the reviewer curious as to what would happen if we were to replace the SQL agent with a RAG agent instead? We suspect a RAG agent will also face similar threats, as it is also a data processing agent that, instead of using precise SQL queries over structured data, ranks documents using algorithms (e.g. k-nearest neighbours) to fetch that data. We focused on the SQL agent because SQL databases are quite common in enterprise settings. The [StackOverflow survey in 2020](https://survey.stackoverflow.co/2020#technology-databases) demonstrated that the top 4 database technologies were all SQL-based (MySQL, PostgreSQL, MicrosoftSQLServer and SQLite), which SQL agents such as [Microsoft SQLServerAgent](https://learn.microsoft.com/en-us/ssms/agent/sql-server-agent) and [Amazon Bedrock Agents](https://aws.amazon.com/blogs/machine-learning/dynamic-text-to-sql-for-enterprise-workloads-with-amazon-bedrock-agents/) can interact with.
> If we misunderstood your question, please feel free to clarify, and we will follow up eagerly.
>
> W3 and Q2:
>
> We agree with the reviewer that adding a list of actionable items/summarised lessons would strengthen our paper. Here are some:
> - Access control is insufficient as a safeguard measure if data entry itself is not adequately sanitised and monitored
> - Adding a monitor or filter at every step of the agentic system, i.e. before and after each user-agent interaction and inter-agent communication, would help flag any malicious activity
> - Enabling monitors or agents to alert a human to suspicious activity immediately is another defensive measure we’d recommend
>
> We have added this to the paper in the Conclusion section (it’s in blue).

---

### Note · Program_Chairs · 2026-01-17
**Submission Desk Rejected by Program Chairs**

The following references in this submission do not refer to real documents and/or have major errors in bibliographic information:

 1. Terry Zhuo, Nelson F Liu, Yizhong Wang, Simin Jiang, Tianyi Zhang, Mohit Iyyer, Luke Zettlemoyer, and Daniel Khashabi. Red teaming language models with language models. In Advances in Neural Information Processing Systems, 2023.
The real paper is published in EMNLP with different authors (refer to https://aclanthology.org/2022.emnlp-main.225.pdf)
2. Xinyue Yang, Shuo Chen, Yiming Zhang, Weiqiang Shi, Yizheng Yu, Qian Li, Shangqing Zhao, Tong Guo, and Hongtao Yu. Jailbreaking chatgpt via prompt engineering: An empirical study. In IEEE Symposium on Security and Privacy, 2024.